# The Effects of Percutaneous Coronary Intervention on the Flow in Acute Coronary Syndrome Patients—Geometry in Focus

**DOI:** 10.3390/jpm12081264

**Published:** 2022-07-31

**Authors:** Agnes Orsolya Racz, Ildiko Racz, Gabor Tamas Szabo, Aron Uveges, Zsolt Koszegi, Bence Penczu, Rudolf Kolozsvari

**Affiliations:** 1Department of Cardiology and Heart Surgery, University of Debrecen Faculty of Medicine, Moricz Zs. Krt 22, Debrecen 4032, Hungary; raczagi21@gmail.com (A.O.R.); iracz73@gmail.com (I.R.); nszgt@med.unideb.hu (G.T.S.); uveges.aron@med.unideb.hu (A.U.); koszegi@med.unideb.hu (Z.K.); 23rd Department of Internal Medicine, Szabolcs-Szatmar-Bereg County Hospital, Nyíregyháza 4400, Hungary; 3Department of Cardiology, Borsod-Abaúj-Zemplén County Hospital, Miskolc 3526, Hungary; penczu.bence@gmail.com

**Keywords:** TIMI frame count, ACS, 2/3D parameters, STEMI, NSTEMI, coronary flow

## Abstract

Evaluation of the effect of three dimensional (3D) coronary plaque characteristics derived from two dimensional (2D) invasive angiography images (ICA) on coronary flow determined by TIMI frame count (TFC) in acute coronary syndrome (ACS) has not been thoroughly investigated. A total of 71 patients with STEMI, and 73 with NSTEMI were enrolled after primary angioplasty. Pre- and post-PCI TFCs were obtained. From 2D images, 3D reconstruction was performed of the culprit vessel, and multiple plaque parameters were measured. In STEMI, the average post-PCI frame count decreased significantly, resulting in better flow. With regards to 2/3D parameters, no differences were found between the STEMI and NSTEMI groups. The 3D parameters in the subgroup with an increase with at least three frames resulting in worsening post-PCI flow were compared to parameters of the patients with improved or significantly not change flow (delta frame count < 3), and greater minimal luminal diameter and area was found in the worsening (increased) frame group. In STEMI 2/3D, parameters showed no correlation with worsening flow, whereas in NSTEMI, greater minimal luminal diameter and area correlated with decreased flow. We can conclude that certain 2/3D parameters can predict slower flow in ACS, resulting in the use of GP IIb/IIIa receptor blocker.

## 1. Introduction

Atherosclerotic heart disease and acute coronary syndrome (ACS), including unstable angina, non-ST-segment elevation (NSTEMI), and ST-segment elevation myocardial infarction (STEMI), are still the leading causes of mortality in Europe and North America [1]. In coronary arteries, plaque formation starts with positive remodeling, or outward growth; the plaque does not usually protrude into the lumen until its volume reaches at least 40%, without resulting in hemodynamic changes. When comparing positive and negative remodeling plaques with similar geometrical appearance, the earlier is more prone to rupture, leading to ACS, being a strong risk factor for further cardiac events [2,3,4]. Based on autopsy findings, Virmani et al. observed that these vulnerable plaques are lipid-rich, with a necrotic core and thin fibrotic cap [5]. In STEMI, the cause of thrombus formation is usually plaque rupture, erosion, or calcified nodule; these, along with lipid rich yellow content of the plaque, can be detected by intravascular ultrasound (IVUS) and optical coherence tomography [6]. In NSTEMI, plaque erosion is the leading cause of the pathology leading to partial or complete vascular occlusion [7]. 

Percutaneous coronary intervention (PCI) and stent implantation are considered the most appropriate approach in ACS [8]. When it comes to 3D changes in the coronary arteries during PCI, pre-stenting arch-cord ratio is an independent risk factor for in-stent restenosis, along with existing causative findings between changes at stent edges, shear stress, and consequential intimal proliferation [9]. In our previous paper on 3D reconstruction from 2D invasive angiography images, percent area stenosis and plaque volume correlated well with invasive fractional flow measurements [10].

With regards to functional evaluation, TIMI (thrombolysis in myocardial infarction) flow and TIMI frame count (TFC) are valuable tools [11]. The occurrence of TIMI 0-2 in ACS is around 15–26% [12]. French et al. stated that the post-PCI 3-week TFC is an independent predictor of five-year survival [13]. With SPECT, as one of the best functional investigation methods in coronary artery disease, perfusion defect size correlated well with pathologically increased TFC or less than TIMI 3 flow [14]. Ample studies investigated the role of coronary thrombus and plaque material embolization, along with both device-oriented (aspiration) and medical prevention (thrombocyte receptor blocking) methods. Farooq et al., hypothesized that certain aspiration method would prevent macroembolization (epicardial artery level), and others might reduce the rate of plaque-caused microembolization and potential slow- or no-reflow (capillary level) [15]. Tanaka et al. proved that coronary microembolization is one of the main causes of slow- or no-reflow phenomena [16]. Jia et al., with the use of IVUS found that the presence of low-attenuation plaques was associated with more than four-fold increase in the risk of TIMI 0–2 [17].

We found little evidence in the literature on how plaque and lumen morphology of the culprit lesion segmented with 2/3D reconstruction from 2D coronary images affects post-PCI coronary flow characteristics in ACS. In the first place, we hypothesized that 2/3D parameters, especially smaller luminal parameters, such as area and diameter, can predict flow changes in ACS. Second, we assumed that the two major forms of ACS, STEMI and NSTEMI, can result in flow difference during PCI due to both owning unique plaque characteristics. Third, both demographic and 2/3D parameters cause greater flow changes, i.e., more than three-frame increase in TFC, which need to be investigated.

## 2. Methods

From our clinical database, 71 patients with STEMI and 73 with NSTEMI were enrolled, who underwent PCI and stent implantation. Demographic data, relevant CV risk factors, and prehospital therapy were collected from the local database (MedSolution, T-Systems, Frankfurt, Germany). Invasive coronary angiography (ICA) images were obtained from GE Innova System (GE Healthcare, Chicago, IL, USA), at the frame rate of 15/sec. Omnipaque 350 mg/mL (GE Healthcare, USA) and Visipaque 300 mg/mL (GE Healthcare, USA) contrast solutions were administered at the flow of 3mL/sec and 6mL/injection.

First, for objective measurement of the coronary flow velocity, TFC was used, being the number of frames counted at the speed of 15/sec, from the point the contrast material entering the culprit vessel until reaching the most distal visible point of the involved coronary artery’s main branch [18]. We excluded patients with TIMI 0-1, since in that case TFC and 3D reconstruction of the involved segment cannot be performed either before or after PCI. Both pre- and post-PCI TFCs were collected.

Second, from the ICA images, using two projections with at least 25-degree apart, the culprit lesion and segment were selected. With QAngioXA 3D (QAngio® XA 3D Research Edition 1.0, Medis Specials bv, Leiden, The Netherlands), 2/3D reconstruction of the involved segments was performed, from the most proximal and distal intact points. From the reconstruction, numerous 2/3D characteristics of the plaque and coronary segment were defined (Figure 1). 

Third, any correlation between TFC and 2/3D parameters was investigated, examining STEMI and NSTEMI subgroups separately. As a cut-off for TFC changes, ≥3 vs. <3 groups were compared based on previous studies investigating the significance level on the number of frame change [19,20,21]. 

## 3. Statistical Methods

The statistical analysis was processed with IBM SPSS 22 software. The data are expressed as the mean ± SD and frequencies and percentages. Continuous variables were evaluated by independent two-tailed t-test or Man–Whitney test. Nominal variables were compared between groups using the chi-squared or Fisher’s exact test, as appropriate. The association between the two groups was studied in logistic regression models, and independent prognostic factors were sought using multiple regression models. *p* values < 0.05 were considered significant.

## 4. Results

When evaluating the demographic data, significant differences between the STEMI and NSTEMI patients were found, having more STEMI patients receiving prehospital sodium-heparin (79 vs. 22%; *p*
**<** 0.001), whereas diabetes (21 vs. 45%; *p* = 0.002), dyslipidemia (18 vs. 56%; *p*
**<** 0.001), and previous myocardial infarction (11 vs. 37%; *p* < 0.001) were more common in NSTEMI (Table 1).

In case of intergroup comparison, we found significantly higher pre-PCI TFC in STEMI patient (32.42 ± 14.49 vs. 28.34 ± 10.57; *p* = 0.056). Evaluating the intragroup results, in NSTEMI, post-PCI TFC decreased, in a non-significant manner (28.34 vs. 26.89; *p* = 0.324), whereas in STEMI, post-PCI TFC decreased significantly (32.42 vs. 24.37; *p* < 0.001). (Table 2/Figure 2)

When looking at 2/3D parameters, there were no significant differences between the groups. Non-significant differences were observed in (not exclusively) 3D obstruction length (18.90 ± 13.64 vs. 20.54 ± 18.79; *p* = 0.551), percent diameter obstruction at minimal lumen diameter (MLD) (57 ± 13.08 vs. 55.88 ± 12.68%; *p* = 0.894), and percent area obstruction at this point (68.83 ± 14.38 vs. 64.47 ± 12.65; *p* = 0.106); in the minimal lumen area (MLA) (1.05 ± 0.55 vs. 1.20 ± 0.62; *p* = 0.126), the obstruction segment volume (58.29 ± 61.27 vs. 77.38 ± 117.05; *p* = 0.224) and in the obstruction segment plaque volume (23.90 ± 27.08 vs. 24.78 ± 33.84; *p* = 0.863). (Table 3)

### Evaluation of TFC Δ ≥ 3 vs. Δ < 3

To investigate whether more prominent TIMI changes might be due to any 2/3D differences or existence of STEMI or NSTEMI, TFC increase with equal or more than 3 (Δ ≥ 3) or less than 3 frames (Δ < 3) groups was created. With regards to demographics, more patients with NSTEMI belonged to group Δ ≥ 3, than to Δ < 3; 12 vs. 59 for STEMI, and 26 vs. 47 for NSTEMI; *p* = 0.010). Moreover, patient receiving sodium-heparin during prehospital treatment belonged more often to the Δ < 3 group (13 vs. 59; *p* = 0.023).

When evaluating the 2/3D parameters in all patients, MLD (1.06 ± 0.311 vs. 0.89 ± 0.29; *p* = 0.002), MLA (1.36 ± 0.66 vs. 1.04 ± 0.54; *p* = 0.003), and reference diameter at MLD (2.21 ± 0.510 vs. 2.17 ± 0.49; *p* = 0.031) were significantly greater in the Δ ≥ 3 group, whereas percent diameter obstruction at MLD (51.23 ± 13.04 vs. 58.8 ± 12.27; *p* = 0.003), percent area obstruction at MLD (59.76 ± 19.05 vs. 69.08 ± 14.29; *p* = 0.002), and area obstruction at MLA (63.42 ± 15.70 vs. 70.99 ± 12.99; *p* = 0.004) were grater in the Δ < 3 group (Table 4).

In STEMI patients, no difference in demographics was observed comparing the Δ ≥ 3 and Δ < 3 groups. With regards to 2/3D parameters, also no significant differences were found between the two subgroups (Table 5).

In NSTEMI patients, no difference was observed in demographics comparing the Δ ≥ 3 or Δ < 3 groups, either. With regards to 2/3D parameters, we found MLD (1.07 ± 0.32 vs. 0.87 ± 0.24; *p* = 0.007) and MLA (1.49 ± 0.68 vs. 1.04 ± 0.52; *p* = 0.002) greater in the Δ ≥ 3 group, whereas for percent diameter obstruction at MLD (50.57 ± 14.11 vs. 58.80 ± 10.85; *p* =0.007), percent area obstruction at MLD (55.52 ± 20.31 vs. 69.42 ± 13.68; *p* = 0.001), and area obstruction at MLA (60.66 ± 16.55 vs. 71.29 ± 12.61; *p* = 0.003), values were higher in the Δ < 3 groups (Table 6).

With regards to odds ratio, lower MLD (OR 14.109, CI 2.185-91.093; *p* = 0.005) and MLA (OR 3.461, CI 1.470-8.150; *p* = 0.004) proved higher risk for getting into Δ ≥ 3, whereas percent diameter obstruction at MLD (OR 0.946, CI 0.907-0.987; *p* = 0.010), percent area obstruction at MLD (OR 0.952, CI 0.922-0.983; *p* = 0.003), and area obstruction at MLA (OR 0.950, CI 0.916-0.986; *p* = 0.006) provided lower risk for Δ ≥ 3. Percent area obstruction at MLD became the strongest independent prognostic factor.

When it comes to in-hospital mortality, only one patient died before discharge in the NSTEMI group. In the STEMI group, this figure was three. When it comes to long-term all-cause death, there is no significant difference between STEMI and NSTEMI patients all through the 6-year follow-up (*p* = 0.789). When it comes to mortality in all patients vs. IIb/IIIa inhibitor receivers, even though there was no significant difference (*p* = 0.321), there is a trend of better survival in the inhibitor group. Moreover, worth mentioning that in the STEMI group, the trend was the same (*p* = 0.170), and no patient receiving IIb/IIIa inhibitor died during the 6-year follow-up.

## 5. Discussion

Visual evaluation during PCI is still considered being the “gold standard” in decision-making to proceed to intervention. Recently, 3D imaging has been improving dramatically, both in anatomical and functional aspects (CT-FFR/QFR). These improvements led to more sophisticated reconstruction software using 2D information to create 3D images. In our latest investigation, we decided to use the 2/3D reconstruction to explain flow changes in STEMI and NSTEMI patients during PCI and to examine, which subset of patients might need supplemental therapy for flow improvement, either device or pharmacological, i.e., administration of IIb/IIIa receptor inhibitor agent, of which indication has become controversial recently.

In our previous paper, we proved that in stable coronary disease patients, 2/3D parameters correlate well with FFR [10]. Umman et al. showed a strong correlation between TFC and FFR, proving post-PCI decrease of TFC and increase of FFR support each other toward successful revascularization [10,14,22]. Hayıroğlu et al. investigated whether post-intervention TFC change could affect the outcome in cardiogenic shock patients with STEMI and found TFC Δ ≥ 2 to be an independent predictor of mortality [23]. Considering the studies above, we can conclude that functional parameters, such as FFR and TFC, correlate well with several 2/3D parameters and mortality.

We developed a methodical investigation of both morphological and functional aspects of hemodynamics in ACS patients with the use of 2/3D segmentation software. When it comes to interpretation of results, the larger pre-PCI TFC value in STEMI patient was probably due to higher dispensed thrombus burden already filling or hardly passing through the capillaries. With regards to post-PCI TFC, there were no differences between the two groups, being independent from the nature of the ACS. In STEMI, the flow improved significantly, which can be interpreted as a higher pre-PCI TFC. In NSTEMI, there was also a trend toward improving coronary flow, but in a non-significant manner. Even though there was no significant difference in the post-PCI TFCs, in NSTEMI it was higher, probably due to more debris from the plaque obliterating the capillaries. Higher number of previous history of diabetes, dyslipidemia, and infarction in NSTEMI can explain this notion.

After 2/3D reconstruction, we found no parameters significantly differ when comparing STEMI and NSTEMI patients, which refers to fairly similar plaque morphological characteristics, giving partial answers to our query with regards to plaque characteristics in different forms of ACS.

When exploring the two groups with an increase in TFC with at least three frames, NSTEMI was a prognostic factor for worsening post-PCI flow, whereas administration of sodium-heparin provided better chances for better flow. With regards to plaque structural parameters, worsening post-PCI flow was mainly due to greater minimal luminal diameter and area. It might be a paradoxon, why greater lumen, hence smaller possible plaque volume causes decreased flow, but it might be due to plaques with multiple, hemodynamically non-significant lesions providing more debris compared to plaques with one more severe, but “lonely” lesion.

All the above findings raise the question, what can be done in a subset of patients with these parameters to improve post-PCI flow. A potential solution for the decreased coronary flow problem during PCI is the administration of thrombocyte GP IIb/IIIa receptor inhibitor. In the last two decades, the role of GP IIb/IIIa receptor inhibitors in ACS have changed from “must to might”. With regards to NSTEMI, Tricoci et al. found that upstream use provided a significant but modest ischemic benefit compared with placebo and proved a trend toward less ischemic event when compared to selective downstream administration [24]. Sciahbasi et al. also declared in their metanalysis that early administration of GP IIb/IIIa receptor inhibitors in NSTEMI at 30 days was associated with significant reduction in ischemic events, to be applied in high ischemic but low bleeding risk patients [25,26].

With regards to STEMI and GP IIb/IIIa receptor inhibitors, de Luca et al. found in their metanalysis in the 30-day mortality that the agent has strong recommendation in high-risk patients [27]. In another meta-analysis, they also proved that early administration of GP IIb/IIIa receptor inhibitors in STEMI and PCI was associated with significant benefits in vascular endpoints and ST-segment resolution. Nevertheless, there was no significant mortality improvement, except for abciximab [28].

Based on the mortality figures, in-hospital mortality was very low, which can be explained by the fact that no patients arrived in cardiogenic shock. When it comes to long-term all-cause death, no significant difference between STEMI and NSTEMI was experienced, which goes parallel with the literature [29]. Moreover, we assume similar pathomechanism in the form of ACS after the occluded artery became opened. With the use of IIb/IIIa receptor inhibitor, there was a trend of better survival in the inhibitor group for all patient groups, which definitely proves at least non-inferiority and effective use. Last, but not least, having no death at 6th year in the STEMI group for patients receiving IIb/IIIa receptor inhibitor indicates safe use.

When it comes to cost effectiveness, Glaser et al. declared that upstream use of GP IIb/IIIa receptor inhibitor was superior to selective use: an 18,000 dollars per year of life gained. When TIMI risk score was applied for stratification, upstream use was only cost effective in those patients with moderate or high risk—a subset of patients where the upstream strategy should be used [30].

Finally, as of the latest guidelines on ACS coronary revascularization, GP IIb/IIIa receptor inhibitors can be considered as a “bail-out” therapy in cases of large thrombus-burden and in slow/no-reflow situation, randomized trials await [8,31,32].

## 6. Conclusion

Several 2/3D parameters obtained in a few minutes with an individualized software at the site of intervention along with measuring TFC can provide a good estimate of the success of revascularization. Even before the actual PCI quick reconstruction of the culprit segment, along with planning the accurate size of stent, one can anticipate the chance of post-PCI low coronary flow. Having this information in hand, the notion of administering GP IIb/IIIa receptor inhibitors, or the use of embolic protection stent or devices can arise.

Taking the above into consideration, patient admitted with NSTEMI, with more than 3-frame increase in post-PCI TFC along with greater minimal luminal diameter and area with 2/3D reconstruction are the “ideal” patient for GP IIb/IIIa receptor inhibitor administration to acquire best potential post-PCI flow. With regards to mortality results, we can conclude:In ACS with opened culprit arteries, the flow characteristics are similar between the two major forms, referring to a similar patomechanism after the vessel is open.Irrespective to the form of ACS, receiving IIb/IIIa receptor inhibitor provided better chance for survival during the six-year follow-up.Last, but not least, no STEMI patient died during the follow-up who received IIb/IIIa receptor inhibitor, proving strong potential benefit in this patient cohort.These findings prove that using IIb/IIIa inhibitor, in select cases, was safe and effective. For more accurate results and predictions in mortality, alternatively a retroprospective analysis with fitted patient cohorts (same 2/3D parameters and TFC changes, naive or GP IIb/IIIa receptor inhibitor administered) would be of high importance.

## 7. Limitations

Excluding patients with TIMI 0-1flow from the investigation caused partly half of ACS patients to stay out of the investigation—our results cannot be implicated in their cases. Possible confounding factors for TFC evaluation are heart rate and the different phases of the heart cycle when the contrast material is injected, and the amount of nitrate given (21). Taking into consideration that from the start to end point, more than one heart cycle might evolve, applying an empiric evaluation, we assume that the difference (or, in this case, possible bias) in the time of contrast material injection, can even out at a certain “high” number of enrolled patients. With regards to nitrate administration, in our lab, we usually administer 200 micrograms of intracoronary nitrate only when severe flow reduction occurs. Heart rate standardization did not happen.

## 8. Ethical considerations

All activities, with regards to creating the document, data collection, and evaluation, have been performed in accordance with the ethical standards laid down in the 1964 Declaration of Helsinki and its later amendments. All patient data, and the right to use them by the permission of the patients, were gathered by the rules and regulations of the local ethical committee. No personal data, applicable for identification of patient, have been used in the document.

## Figures and Tables

**Figure 1 jpm-12-01264-f001:**
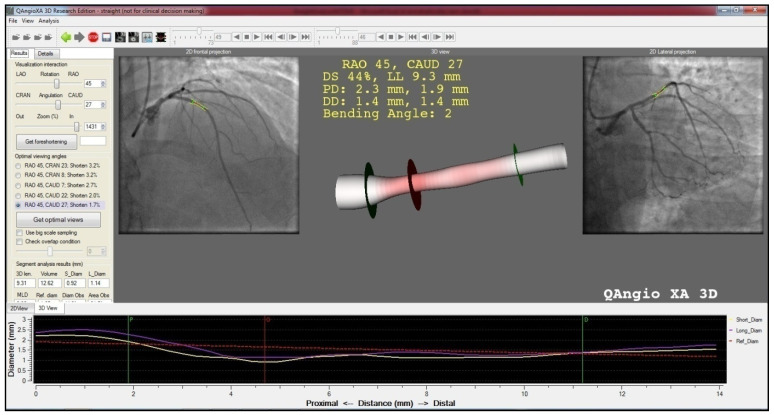
2D/3D reconstruction image of the plaque and the involved coronary segment with the use of QAngioXA 3D. The left and right discs represent the starting and end point of reconstruction, whereas the middle disc shows the narrowest point of the segment.

**Figure 2 jpm-12-01264-f002:**
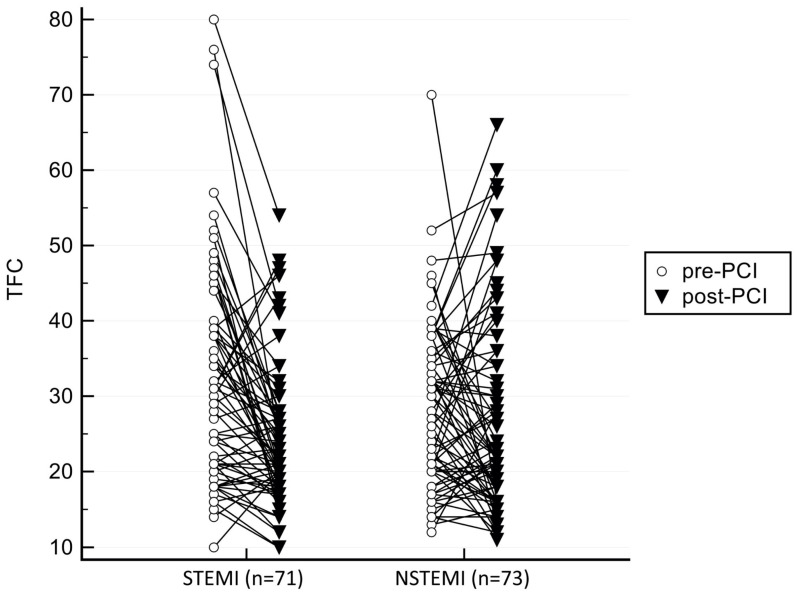
Intragroup pre- and post-PCI TFCs, both in STEMI and NSTEMI, earlier group showing significant changes.

**Table 1 jpm-12-01264-t001:** Demographic data comparing STEMI and NSTEMI patients. Receiving sodium-heparin in ambulance being more common in the earlier, whereas diabetes, dyslipidemia, and previous myocardial infarction in the latter.

	STEMI (n = 71)	NSTEMI (n = 73)	*p*
Age	64.90 ± 10.80	65.10 ± 9.99	0.910
Gender			
-male	46 (65%)	51 (70%)	0.516
-female	25 (35%)	22 (30%)	
Hypertension	39 (55%)	51 (70%)	0.064
**Diabetes**	**15 (21%)**	**33 (45%)**	**0.002**
**Dyslipidemia**	**13 (18%)**	**41 (56%)**	**< 0.001**
Smoking	36 (51%)	35 (48%)	0.741
**Earlier infarction**	**8 (11%)**	**27 (37%)**	**< 0.001**
**Sodium-heparin in ambulance**	**56 (79%)**	**16 (22%)**	**< 0.001**

**Table 2 jpm-12-01264-t002:** Intra- and intergroup comparison of pre- and post-PCI TFCs, both in STEMI and NSTEMI patient.

	STEMI (n = 71)	NSTEMI (n = 73)	*p*
TFC pre-PCI	32.42 ± 14.49	28.34 ± 10.57	0.056
TFC post-PCI	24.37 ± 9.21	26.89 ± 13.16	0.184
Significance of ΔTFC	*p* < 0.001	*p* = 0.324	

**Table 3 jpm-12-01264-t003:** 2/3D parameters; no significant differences evolved.

	STEMI (n = 71)	NSTEMI (n = 73)	*p*
3D obstruction length (mm)	18.90 ± 13.64	20.54 ± 18.79	0.551
Minimal lumen diameter (mm)	0.92 ± 0.32	0.95 ± 0.29	0.561
Reference diameter at MLD (mm)	2.17 ± 0.47	2.20 ± 0.52	0.726
Percent diameter obstruction at MLD (%)	56,99 ± 13.05	55.87 ± 12.65	0.602
Percent area obstruction at MLD (%)	68.83 ± 14.38	64.47 ± 12.65	0.106
Minimal lumen area (mm^2^)	1.05 ± 0.55	1.20 ± 062	0.126
Reference area at MLA (mm^2^)	3.81 ± 1.72	4.02 ± 1.20	0.484
Area obstruction at MLA (%)	70.52 ± 13.3	67.50 ± 14.94	0.201
Obstruction segment mean diameter (mm)	1.79 ± 0.35	1.89 ± 0.42	0.127
Obstruction segment volume (mm^3^)	58.29 ± 61.27	77.38 ± 117.05	0.224
Obstruction segment plaque volume (mm^3^)	23.90 ± 27.08	24.78 ± 33.84	0.863
Obstruction segment reference volume (mm^3^)	79.32 ± 81.93	97.72 ± 141.50	0.343
Obstruction segment mean reference diameter (mm)	2.17 ± 0.45	2.21 ± 0.52	0.624
Arch-chord ratio	1.07 ± 0.09	1.10 ± 0.19	0.180
Lesion eccentricity index	0.27 ± 0.15	0.23 ± 0.12	0.106

**Table 4 jpm-12-01264-t004:** 2/3D parameters when patients are separated into post-PCI TFC changes Δ ≥ 3 or Δ < 3.

	Δ ≥ 3 (n = 38)	Δ < 3 (n = 106)	*p*
3D obstruction length (mm)	18.98 ± 19.45	20.00 ± 15.28	0.744
**Minimal lumen diameter (MLD) (mm)**	**1.06 ± 0.311**	**0.89 ± 0.29**	**0.002**
**Reference diameter at MLD (mm)**	**2.21 ± 0.510**	**2.17 ± 0.49**	**0.031**
**Percent diameter obstruction at MLD (%)**	**51.23 ± 13.04**	**58.8 ± 12.27**	**0.003**
**Percent area obstruction at MLD (%)**	**59.76 ± 19.05**	**69.08 ± 14.29**	**0.002**
**Minimal lumen area (mm^2^)**	**1.36 ± 0.66**	**1.04 ± 0.54**	**0.003**
Reference area at MLA (mm^2^)	4.05 ± 1.86	3.87 ± 1,87	0.617
**Area obstruction at MLA (%)**	**63.42 ± 15.70**	**70.99 ± 12.99**	**0.004**
Obstruction segment mean diameter (mm)	1.91 ± 0.40	1.82 ± 0.39	0.220
Obstruction segment volume (mm^3^)	73.89 ± 131.56	65.84 ± 76.85	0.652
Obstruction segment plaque volume (mm^3^)	21.01 ± 27.33	25.54 ± 31.72	0.435
Obstruction segment reference volume (mm^3^)	90.88 ± 150.34	87.84 ± 101.75	0.890
Obstruction segment mean reference diameter (mm)	2.20 ± 0.47	2.19 ± 0.49	0.868
Arch-chord ratio	1.10 ± 0.17	1.08 ± 0.14	0.553
Lesion eccentricity index	0.23 ± 0.13	0.26 ± 0.14	0.331

**Table 5 jpm-12-01264-t005:** 2/3D parameters in STEMI patients comparing TFC changes between Δ ≥ 3 and Δ < 3 groups.

	Δ ≥ 3 (n = 12)	Δ < 3 (n = 59)	*p*
3D obstruction length (mm)	14.18 ± 6.69	19.86 ± 14.51	0.421
Minimal lumen diameter (mm)	1.01 ± 0.29	0.90 ± 0.32	0.262
Reference diameter at MLD (mm)	2.16 ± 0.43	2.17 ± 0.48	0.812
Percent diameter obstruction at MLD (%)	52.67 ± 10.80	57.86 ± 13.38	0.220
Percent area obstruction at MLD (%)	68.93 ± 12.22	68.81 ± 14.87	0.794
Minimal lumen area (mm^2^)	1.09 ± 0.54	1.04 ± 0.56	0.667
Reference area at MLA (mm^2^)	3.80 ± 1.44	3.81 ± 1.78	0.724
Area obstruction at MLA (%)	69.39 ± 12.22	70.74 ± 13.39	0.629
Obstruction segment mean diameter (mm)	1.78 ± 0.27	1.79 ± 0.37	0.896
Obstruction segment volume (mm^3^)	37.87 ± 19.25	62.45 ± 66.01	0.560
Obstruction segment plaque volume (mm^3^)	18.11 ± 13.79	25.07 ± 29.00	0.471
Obstruction segment reference volume (mm^3^)	54.22 ± 30.65	84.42 ± 88.13	0.443
Obstruction segment mean reference diameter (mm)	2.15 ± 0.36	2.18 ± 0.47	0.939
Arch-chord ratio	1.05 ± 0.05	1.07 ± 0.10	0.454
Lesion eccentricity index	0.31 ± 0.14	0.26 ± 0.16	0.200

**Table 6 jpm-12-01264-t006:** 2/3D parameters in NSTEMI patients comparing TFC changes between Δ ≥ 3 and Δ < 3 groups.

	Δ ≥ 3 (n = 26)	Δ < 3 (n = 47)	*p*
3D obstruction length (mm)	21.19 ± 22.89	20.17 ± 10.35	0.826
**Minimal lumen diameter (MLD) (mm)**	**1.07 ± 0.32**	**0.87 ± 0.24**	**0.007**
Reference diameter at MLD (mm)	2.24 ± 0.55	2.17 ± 0.50	0.600
**Percent diameter obstruction at MLD (%)**	**50.57 ± 14.11**	**58.80 ± 10.85**	**0.007**
**Percent area obstruction at MLD (%)**	**55.52 ± 20.31**	**69.42 ± 13.68**	**0.001**
**Minimal lumen area (mm^2^)**	**1.49 ± 0.68**	**1.04 ± 0.52**	**0.002**
Reference area at MLA (mm^2^)	4.16 ± 2.03	3.95 ± 2.00	0.663
**Area obstruction at MLA (%)**	**60.66 ± 16.55**	**71.29 ± 12.61**	**0.003**
Obstruction segment mean diameter (mm)	1.97 ± 0.44	1.85 ± 0.41	0.250
Obstruction segment volume (mm^3^)	90.51 ± 156.66	70.11 ± 89.19	0.480
Obstruction segment plaque volume (mm^3^)	22.35 ± 31.87	26.13 ± 35.14	0.651
Obstruction segment reference volume (mm^3^)	107.80 ± 179.15	92.14 ± 117.50	0.654
Obstruction segment mean reference diameter (mm)	2.23 ± 0.51	2.20 ± 0.53	0.851
Arch-chord ratio	1.12 ± 0.20	1.09 ± 0.18	0.508
Lesion eccentricity index	0.20 ± 0.10	0.25 ± 0.13	0.064

## Data Availability

Not applicable.

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
