# Peer review of "The Effects of Percutaneous Coronary Intervention on the Flow in Acute Coronary Syndrome Patients—Geometry in Focus"

_jpm, 2022, doi:10.3390/jpm12081264_

Round 1

Reviewer 1 Report

The Authors present a small series of angio 2/3D ACS analysis to predict the TFC after PCI. 

There are some methodological problems . First of all STEMI patients, frequently have slow o no time flow at presentation, differently from NSTEMI patients. So, NSTEMI can not be considered a risk factor for worse TFC.

Secondly, NSTEMI and STEMI patients have a completely different risk of no reflow because of the time of PCI and systemic thrombotic activation.

Moreover, in the imaging era (particularly OCT), angioguided analysis for plaque morphology can be considered accurate.

Finally, the Authors suggest gpIIb/IIIa inibithors use upstream in  some settings. This is the opposite of RCT and guidelines suggest

Author Response

Dear Reviewer 1,

Thank you for your comment, critics, and suggestions. Answers as follow:

  1. First of all STEMI patients, frequently have slow o no time flow at presentation, differently from NSTEMI patients. So, NSTEMI can not be considered a risk factor for worse TFC.

Yes, indeed, in STEMI patient in appr. 40-50 % of the cases occlusion persists when angiography is done. One limitation of our research is that for the 2/3D and the pre/post TFC measurements, we needed at least TIMI 2 flow pre-PCI. Thus, a certain number of STEMI patients do not fit into our method, but still a quite big number of STEMI patients were  to be enrolled.

  1. Secondly, NSTEMI and STEMI patients have a completely different risk of no reflow because of the time of PCI and systemic thrombotic activation.

As stated above, we enrolled patient from both sides only if flow was present in the culprit artery. The extensive prothrombogenic and platelet activating substance from the ruptured plaque in STEMI is partially washed out by the time patient reaches the catheterization lab and the measurements can be performed. In that sense, the thrombogenicity of the culprit lesions in STEMI and NSTEMI can have partially same characteristics.

For the benefit of flow, in STEMI the iv. sodium-heparin given by the paramedics, is accountable for the flow improvements. On the NSTEMI side, if the procedure is not an urgent one, LMWH and thrombocyte inhibitors are on the pro-flow side after a longer period from the diagnosis to revascularization. Even though, anticoagulation and thrombocyte activation systems interfere in their pathways, it would be very hard to compare the two ways of coagulation patterns. Based on the above, we considered these factors to possibly even out in both patient groups by the time of intervention. We calculated the 6-year mortality comparing the STEMI and NSTEMI group, and there was no significant difference, which can be interpreted as fairly similar patomechanism after the artery is open.

  1. Moreover, in the imaging era (particularly OCT), angioguided analysis for plaque morphology can be considered accurate.

That is absolutely correct, but the local circumstances always have positive or negative effect on the technique/method one uses. Our catheterization lab is the highest provider level in the region, having OCT on board, but still not using it due to financial (catheter price) and local reasons (only few of the staff can interpret the results properly). Thus, a reconstruction software easily available and at moderate cost, that can be used for the same problem, even if not with the same efficacy, can be of good use in labs with limited access to state-of-the-art techniques.

  1. Finally, the Authors suggest gpIIb/IIIa inibithors use upstream in some settings. This is the opposite of RCT and guidelines suggest.

The reviewer states correctly, quote:

  1. „Nevertheless, use should be considered for bail-out situations or thrombotic complications and may be considered for high-risk PCI in patients without pre-treatment with P2Y12 receptor inhibitors” (ESC NSTEMI, 2020)
  2. „Using GP IIb/IIIa inhibitors as bailout therapy in the event of angiographic evidence of a large thrombus, slow- or no-reflow, and other thrombotic complications is reasonable, although this strategy has not been tested in a randomized trial. Overall, there is no evidence to recommend the routine use of GP IIb/IIIa inhibitors for primary PCI.” (ESC Guidelines for the management of acute myocardial infarction in patients presenting with ST-segment elevation, 2017)
  3. „Thus, GP IIb/IIIa inhibitors may only be considered in specific ‘bail-out’ situations including high intraprocedural thrombus burden, slow flow, or no-flow with closure of the stented coronary vessel.” (ESC/EACTS Guidelines on Myocardial Revascularization, 2018)

Based on the above, we can conclude, that GP IIb/IIIa inhibitors should be used in the existing evidence of high thrombus burden, in high-risk patients, or with significant flow insufficiency, especially when no pre-treatment with P2Y12 receptor inhibitors available, usually in a bail-out manner. Should we able to predict further flow decrease post-stent implantation from 2/3D parameters, or using TFC and the reconstruction information together, it could give us a good prediction for the usefulness of GP IIb/IIIa- upstream of bailout. This situation resembles that of thrombus aspiration and IABP history. Both used very often in certain situations many years ago, but recently being both considered to apply in some „special” cases. All three, device or pharmacological intervention can and should be used in a few types of cases- but those would benefit from them with high probability. Finally, guidelines usually state as a final comment, that interventionalist have the right to make the decision at their own discretion in uncertain cases- anything, that helps the physician to make the best possible decision, should be considered.

Reviewer 2 Report

I have reviewed the manuscript entitled 'The effects of percutaneous coronary intervention on the flow in acute coronary syndrome patients- geometry in focus'. 

The article appears to be novel however several issues should be addressed 

First the endpoints should be given as in-hospital mortality and long-term mortality as these patients have admitted to hospital with MI. The contribution of the study is very limited otherwise. Is there an effect of this calculations on the endpoints.?

The role of TIMI blood flow both pre-procedural and post-procedural are very important in the endpoints of the patients. Please mention it in the discussion citing 'Predictors of In-Hospital Mortality in Patients With ST-Segment Elevation Myocardial Infarction Complicated With Cardiogenic Shock'.

Author Response

Dear Reviewer 2,

Thank you for your valuable comments/questions/suggestions. Answers as follow:

1. First the endpoints should be given as in-hospital mortality and long-term mortality as these patients have admitted to hospital with MI. The contribution of the study is very limited otherwise. Is there an effect of this calculations on the endpoints.?

All-cause mortality is one of the most important hard endpoints in a study and can give power to the novel findings.

With regards to in-hospital mortality, in the NSTEMI group, there was only one patient involved. In the STEMI group, this figure was three. These are very low numbers, which can be explained by the fact that no patients arrived in cardiogenic shock.

When it comes to long-tern all-cause death, there is no significant difference between STEMI and NSTEMI patients all through the six-year follow-up (p=0,789), which goes parallel with the literature (Bouisset F, RuidavetsJB, Dallongeville J, Moitry M, M Montaye M, Biasch K, Ferrières J. Comparison of Short- and Long-Term Prognosis between ST-Elevation and Non-ST-Elevation Myocardial Infarction  Clin Med. 2021 Jan 7;10(2):180.) When it comes to mortality with regards to all patient vs. IIb/IIIa receptor inhibitor receivers, even though there is no significant difference (p=0,321), there is a trend of better survival in the inhibitor group. Also, worth mentioning, that in the STEMI group, the trend was the same (P=0,170), and no patient receiving IIb/IIIa receptor inhibitor died during the follow up. Taking the above into consideration, we can conclude:

  1. When it comes to in ACS with opened culprit arteries, there is no different flow characteristic between the two major forms, referring to a similar patomechanism after the vessel is open.
  2. Irrespective to the form of ACS, whoever received IIb/IIIa receptor inhibitor, had better chance for survival in the six-year follow-up
  3. Last, but not least no STEMI patient died during the follow-up who received IIb/IIIa receptor inhibitor.

These findings prove that using IIb/IIIa receptor inhibitor in select cases was safe and effective.

2. The role of TIMI blood flow both pre-procedural and post-procedural are very important in the endpoints of the patients. Please mention it in the discussion citing 'Predictors of In-Hospital Mortality in Patients With ST-Segment Elevation Myocardial Infarction Complicated With Cardiogenic Shock'.

Thank you for mentioning the article, which clearly shows, that in high-risk cardiogenic patients with STEMI TFC D³2 is an independent prognostic factor of mortality, which highlights the importance of optimal TFC not just pre-PCI, but also post-PCI. This finding provides evidence, that all efforts should be made to have the best flow in the coronary in ACS. The reference is cited in the discussion part.

Round 2

Reviewer 1 Report

nice job

Reviewer 2 Report

Thank you for the required revisions.